# The Metabolic Rearrangements of Bariatric Surgery: Focus on Orexin-A and the Adiponectin System

**DOI:** 10.3390/jcm9103327

**Published:** 2020-10-16

**Authors:** Anna Valenzano, Nicola Tartaglia, Antonio Ambrosi, Domenico Tafuri, Marcellino Monda, Antonietta Messina, Francesco Sessa, Angelo Campanozzi, Vincenzo Monda, Giuseppe Cibelli, Giovanni Messina, Rita Polito

**Affiliations:** 1Department of Clinical and Experimental Medicine, University of Foggia, 71100 Foggia, Italy; anna.valenzano@unifg.it (A.V.); francesco.sessa@unifg.it (F.S.); giuseppe.cibelli@unifg.it (G.C.); 2General Surgery, Department of Medical and Surgical Sciences, University of Foggia, 71100 Foggia, Italy; nicola.tartaglia@unifg.it (N.T.); antonio.ambrosi@unifg.it (A.A.); 3Department of Motor Sciences and Wellness, University of Naples “Parthenope”, 80133 Naples, Italy; domenico.tafuri@uniparthenope.it; 4Department of Experimental Medicine, Section of Human Physiology and Unit of Dietetic and Sport Medicine, Università degli Studi della Campania Luigi Vanvitelli, 80138 Naples, Italy; marcellino.monda@unicampania.it (M.M.); Antonietta.messina@unicampania.it (A.M.); vincenzo.monda@unicampania.it (V.M.); 5Pediatrics, Department of Medical and Surgical Sciences, University of Foggia, 71100 Foggia, Italy; angelo.campanozzi@unifg.it

**Keywords:** obesity, adipose tissue, adiponectin, central nervous system, Orexin-A, bariatric surgery

## Abstract

The accumulation of adipose tissue represents one of the characteristics of obesity, increasing the risk of developing correlated obesity diseases such as cardiovascular disease, type 2 diabetes, cancer, and immune diseases. Visceral adipose tissue accumulation leads to chronic low inflammation inducing an imbalanced adipokine secretion. Among these adipokines, Adiponectin is an important metabolic and inflammatory mediator. It is also known that adipose tissue is influenced by Orexin-A levels, a neuropeptide produced in the lateral hypothalamus. Adiponectin and Orexin-A are strongly decreased in obesity and are associated with metabolic and inflammatory pathways. The aim of this review was to investigate the involvement of the autonomic nervous system focusing on Adiponectin and Orexin-A after bariatric surgery. After bariatric surgery, Adiponectin and Orexin-A levels are strongly increased independently of weight loss showing that hormone increases are also attributable to a rearrangement of metabolic and inflammatory mediators. The restriction of food intake and malabsorption are not sufficient to clarify the clinical effects of bariatric surgery suggesting the involvement of neuro-hormonal feedback loops and also of mediators such as Adiponectin and Orexin-A.

## 1. Introduction

Obesity is characterized by an altered metabolic and inflammatory profile leading to various metabolic, inflammatory, and immune diseases [1]. The accumulation of visceral adipose tissue is a principal characteristic of obesity. It accumulates in the abdominal area of the body and is dangerous for health. Indeed, the endocrine function of adipose tissue is strongly influenced by the presence of visceral adipose tissue. As reported by Xin et al., 2020, in obesity, adipocytes are dysfunctional with an excessive secretion of multiple pro-inflammatory adipokines, contributing to a chronic inflammatory reaction and promoting the progression of metabolic and cardiovascular complications [2]. It is well known that white adipocytes of visceral fat are particularly active in the release of adipokines. Although adipocytes secrete a large variety of bioactive molecules with widespread systemic effects contributing to numerous physiological and pathological processes, the autocrine and paracrine actions of these molecules are highly complex, and our understanding of these processes is likely rudimentary [3]. The complexity of obesity consists of adipose tissue recognized as an endocrine organ producing adipokines, the active protein with pleiotropic functions in the regulation of energy metabolism, insulin sensitivity, inflammation, atherosclerosis, and proliferation. Among these, Adiponectin is the most abundant product of white adipose tissue (WAT). It is produced in various oligomers of different molecular weight and it is negatively correlated with obesity [4].

During obesity development there is also an involvement of the sympathetic system. The central nervous system, through the production of hypotalamic mediators, acts on adipose tissue (AT) regulating its function, both physiologically and patho-physiologically. In particular, through the production of Orexin-A, a hypothalamic peptide, the central nervous system increases the sympathetic stimulation of WAT and thereby increasing lipolysis [5,6,7]. Bariatric surgery is necessary when there is severe obesity and this technique is able to induce an improvement or resolution of many obese related conditions and to improve quality of life, inducing weight loss and a rearrangement of metabolic and hormone pathways. From data in the literature, it is clear that bariatric surgery is capable not only of acting from a mechanical-anatomical point of view, reducing the size of the stomach and therefore the intake of food, but also improving some metabolic parameters such as the production of adipocytokines and hypothalamic peptides. There is a strong metabolic interconnection between the central nervous system, the digestive system and adipose tissue. In light of this evidence, in this review, the functional metabolic changes in the sympathetic and para-sympathetic nervous systems through the production of Adiponectin and Orexin-A induced by bariatric surgery are elucidate.

## 2. Adipose Tissue: General Characteristics

AT is an endocrine organ, composed of adipocytes and pervaded by many innate and adaptative immune cells [8,9,10]. In obesity, the excessive expansion of AT mass induces the recruitment of numerous immune cells leading to an imbalance in adipokine production. AT exerts its metabolic function through the production of adipokines, among these is Adiponectin [11,12,13]. Literature data demonstrated that obesity strongly correlates to immune and autoimmune disease development [11]. The immune system monitors and responds to specific metabolic cues in both pathologic and non-pathologic settings. The immune system continuously communicates with AT. These systems influence each other. In addition, it is well known that the immune system is influenced by environmental changes as well as nutritional factors [11,12]. Imbalanced nutrition strongly influences the function and development of the immune system, depressing it and/or reducing immune competence. As previously reported, AT is pervaded by immune system cells leading to an alteration of pro-inflammatory cytokines such as leptin, TNF-α, L-6, IL8 and anti-inflammatory cytokines such as Adiponectin and IL-10 [11,12,13,14,15].

## 3. Adiponectin: General Characteristics

Adiponectin is an abundant adipokine produced by AT, representing about 0.1% of total serum proteins. This adipokine is present as oligomers of different molecular weight: low molecular weight (LMW), medium molecular weight (MMW) and high molecular weight (HMW) that are the most biologically active [8]. Adiponectin has pleiotropic functions on different target tissues through the presence of its receptors: AdipoR1, AdipoR2 and T-cadherin. The main metabolic functions of Adiponectin are exerted on the liver, muscle, and AT; in fact, the metabolites of Adiponectin affect glucose homeostasis and the metabolism of fatty acids through a primary action at the level of muscles and the liver [9]. It increases insulin sensitivity and reduces hepatic neoglucogenesis, increases glucose uptake by adipocytes and myocytes, increasing the oxidation of free fatty acids in muscles and preventing the increase of free fatty acids and triglycerides as a result of a high fat diet. Numerous studies, both in vitro and in vivo, have also characterized the anti-inflammatory, anti-atherogenic and anti-angiogenic effects of this protein. The anti-inflammatory effects of Adiponectin include the suppression of pro-inflammatory cytokine production, such as TNF-α and IL-6, C-reactive protein and growth factors, and the modulation of the expression of the anti-inflammatory cytokine IL-10 in monocytes and macrophages. On the contrary, TNF-α and other inflammatory markers (IL-6, C-reactive protein, SAA, tPA, MCP-1) and glucocorticoids suppress and regulate Adiponectin production [10] (Figure 1). The anti-atherogenic effects include the modulation of the inflammatory response inhibiting monocyte adhesion and macrophage polarization. In addition, Adiponectin is able to inhibit endothelial cell proliferation and promote apoptosis with a consequent antitumor effect [1]. On the contrary, in cases of vascular damage, Adiponectin regulates the endothelial response to damage by suppressing apoptosis acting on the regulation of proliferation and differentiation of osteoblasts. This adipokine has a pleiotropic function, it is not only a metabolic mediator, but also an inflammatory and immune mediator [11]. As regards its metabolic effects, Adiponectin is involved in glucose and lipid metabolism. It is involved in glucose homeostasis and in the metabolism of fatty acids through a primary action in muscles and the liver by means of AMPK phosphorilation [12]. Furthermore, Adiponectin increases insulin sensitivity and glucose uptake increasing GLUT-4 translocation by adipocytes and myocytes. In the muscle, it increases the oxidation of free fatty acids and prevents the increase of free fatty acids and triglycerides as a result of a high fat diet [12]. Many data in the literature report that Adiponectin negatively correlates with anthropometric parameters such as body mass index (BMI) and body weight, and also with metabolic parameters such as glycemia, total cholesterol, and LDL cholesterol. On the contrary, it is positively correlated with HDL-cholesterol [13]. In addition, several studies report that Adiponectin serum levels are strongly decreased in obese subjects compared to healthy subjects; however, Adiponectin serum levels negatively correlate with the risk of developing obesity related diseases [14] (Figure 1A).

## 4. Orexin-A: General Characteristics

The lateral hypothalamus produces an important neuropeptide; Orexin-A (hypocretin-1) [16]. It plays an important role in peripheral energy balance, suggesting the involvement of central nervous system (CNS) mechanisms, and coordinates sleep-wakefulness and food-seeking, especially in the physiological state of fasting stress [17]. Orexin-A exerts its functions by binding its receptors, Orexin-1 receptor (Ox1) and Orexin-2 receptor (Ox2), two G-protein coupled receptors [18,19,20]. The Orexin system is involved in physiological and pathophysiological processes [21,22,23] (Figure 1B). Many metabolic molecules influence Orexin-A activity; in particular, glucose, leptin, and amino acids and also some environmental factors increase Orexin-A levels during the waking phase of the circadian cycles and fasting or periods of caloric restriction. This neuropeptide is able to regulate physiological and behavioral processes impacting on energy balance and metabolic status, physical activity, blood glucose levels, and food intake [20,21,22]. Overall, it is well known that Orexin-A is involved in the regulation of insulin sensitivity, energy expenditure and metabolic rate. In addition, it regulates immune processes and inflammatory response, in particular, it has an anti-inflammatory action [21,24,25]. Orexin-A exerts its metabolic effects acting on MAPK pathways, through PGC-1α. Data in the literature report that PGC-1α is able to act on neuronal metabolism involving the orexinergic system [20]. It is involved in metabolic pathways and also in different pathologies such as obesity, diabetes, and chronic neurodegenerative diseases [20]. In vitro studies report that Orexin-A, through PGC-1α, activates HIF-1a that may be a link between Orexin and cellular metabolic signaling pathways relevant to obesity [22]. In addition, it is well known that Orexin-A regulates various physiological functions activating phospholipase C/protein kinase C and AC/cAMP/PKA pathways [5,26]. In addition, this neuropeptide exerts its metabolic functions on energy metabolism regulating feeding behavior and energy expenditure [26,27,28,29]. Orexin-A directly acts on AT inducing lipolysis, independently of food intake. As reported by Perez-Leighton et al., the injection of Orexin-A into the lateral thalamus of SD rats for 10 consecutive days reduced diet-induced obesity without affecting food intake [30] (Figure 1B). Previous studies demonstrated that Orexin-A reduces adipogenesis in human intra-abdominal, but not subcutaneous, adipocytes [31,32].

Orexin-A is involved in inflammatory responses as an anti-inflammatory mediator [15]. In obesity, Orexin-A serum levels are strongly reduced and inversely correlate with BMI and with pro-inflammatory mediators such as C-reactive protein and TNF-a. On the contrary, Orexin-A levels positively correlate with Adiponectin serum levels and HDL-cholesterol [33].

## 5. Bariatric Surgery: Why Is It Done and Who Is It for?

The prevalence of overweight and obese people is increasing globally. Recently, many young people (20–30 years old) have been reported to be overweight or obese. Being overweight and obesity are associated with an increased risk of morbidity and mortality [34].

Bariatric surgery is a valid strategy for weight loss. Gastric bypass and other weight loss surgeries are characteristics of bariatric surgery. It is used when diet and physical activity are insufficient for weight loss and/or various serious health problems have caused excessive weight.

The current indications for bariatric surgery refer to the severity of obesity and the potential reversibility of the clinical conditions. The evaluation of BMI is considered a marker for the indication for surgery. In particular, this surgery is performed in subjects with a BMI > 40 kg/m^2^, in the absence of any other comorbidities and with BMI > 35 kg/m^2^ with obesity-associated comorbidities [35,36,37].

Bariatric surgery consists of some procedures limiting food intake and other procedures reducing the body’s ability to absorb nutrients. Malabsorptive bariatric procedures divert the flow of bile and pancreatic enzymes from food and therefore limit the digestion and absorption of nutrients, resulting in reduced calorie intake and subsequent weight loss. Essential micronutrients such as vitamins and trace elements are also absorbed to a lesser extent, potentially leading to severe side effects [35]. In addition, various bariatric surgery techniques use both procedures. Among these, gastric bypass is a reversible technique, decreasing the amount of food intake and also the absorption of nutrients. The most common performed bariatric surgery worldwide is sleeve gastrectomy as reported by Angrisani et al., 2014 [38]. This procedure consists of removing about 80% of the stomach, leaving a long, tube-like pouch. Generally, sleeve gastrectomy induces notable weight loss, because it reduces stomach size and limits food intake. It also produces less of the appetite-regulating hormone ghrelin, which may lessen the desire to eat. In addition, ghrelin reduction is induced by both gastric bypass and sleeve gastrectomy [39]. In addition, an adjustable gastric band (AGB) is another surgery technique. It has advantages including significant weight loss. It is an inflatable silicone device placed around the top portion of the stomach to treat obesity, intended to decrease food consumption [38]. Another procedure is a biliopancreatic diversion with duodenal switch (BPD/DS). It is a complex procedure that tackles weight loss in three different ways. First, a sleeve gastrectomy is performed. For this, a large portion of the stomach is removed with a stapling instrument, leaving a narrow tube, or sleeve, from the top to near the bottom of the stomach. The second part of the procedure reroutes food away from the upper part of the small intestine, which is the natural path of digestion. This cuts back on how many calories and nutrients the body is able to absorb. The small intestine is divided and a connection is made near the end of the small intestine. The third part of the BPD/DS procedure changes the normal way that bile and digestive juices break down food. This cuts back on how many calories are absorbed, causing still more weight loss. One end of the small intestine is connected to the duodenum, near the bottom of the stomach [40]. Bariatric surgery has many beneficial effects, but it has many serious risks and side effects [41]. Moreover, after bariatric surgery, a lifestyle change is necessary through healthy diet and regular physical activity making the beneficial effects of bariatric surgery last longer [36]. Among the beneficial effects of bariatric surgery is knowing that it increases weight loss and reduces the risk of obesity and its associated diseases. In addition, before bariatric surgery, obese subjects are put on a healthy diet with physical activity to induce weight loss. This surgical intervention is carried out on subjects with severe obesity. The current indications for bariatric surgery state that the surgery is necessary when there is severe obesity and the potential reversibility of the clinical conditions. Classically, starting from the Consensus Conference of the American National Institute of Health (1991), BMI is considered to be decisive, but it must be kept in mind that it has important limits; not being in able to highlight the distribution and breakdown of lipid accumulation in the form of somatic or visceral fat, a key factor in determining the metabolic syndrome; and also the different distribution of fat in relation to age, sex and race. For this reason, BMI is considered an important benchmark, but not the only one for establishing the indication for surgery. At the same time, BMI, also considered in its historical dimension as the maximum value reached by the patient, makes it possible to give indications for bariatric surgery. Finally, BMI is evaluated, together with metabolic, functional and psychological parameters, always in an overall balance between risks and benefits, in patients with a BMI > 40 kg/m^2^, in the absence of any other comorbidities; or with a BMI > 35 kg/m^2^, in the presence of comorbidities among those classically considered to be associated with obesity, including type 2 diabetes mellitus (T2DM) resistant to medical treatment [36,37].

As reported by Chang et al., bariatric surgery has substantial and sustained effects on weight and significantly ameliorates obesity-attributable comorbidities in the majority of bariatric surgery patients [35]. The reoperation rate of an adjustable gastric band is higher than that of gastric bypass and sleeve gastrectomy, and the weight loss outcomes of an adjustable gastric band are less substantial than sleeve gastrectomy or gastric bypass [41]. Bariatric surgery is able to induce an improvement or resolution of many obese related conditions and to improve quality of life, also inducing a rearrangement of metabolic and hormone pathways [42]. As reported by Zsombok, there is an improvement in metabolic profile and the remission of type 2 diabetes after bariatric surgery well before weight loss [38]. The author reported that bariatric surgery could alter the neural communication between the gastrointestinal system and the brain, interfering with the autonomic output to the visceral organs, including the liver. In addition, incretins, among these is glucagon-like peptide 1 (GLP-1), are able to influence the central nervous system. Data in the literature reported that the level of GLP-1 is significantly increased after bariatric surgery and could have a key anti-diabetic effect, regardless of weight loss [42]. Moreover, bariatric surgery, the central nervous system and AT influence each other through the production of neuropeptide such as Orexin-A and adipocytokines such as Adiponectin, as suggested by several studies [6].

## 6. Effects of Bariatric Surgery on Adiponectin and Orexin-A

It is well known that the beneficial effects of bariatric surgery have shifted from the contribution of simple gastric diversion and restriction to an energetic pursuit of the contribution of gastrointestinal hormones, secretions, and the microbiome [43]. There is an involvement of sympathetic and parasympathetic nervous systems in bariatric surgery. Several studies reported that the automatic nervous system and parasympathetic tone are involved in the regulation of inflammation. In addition, it is associated with hypothalamus-pituitary-adrenal axis (HPA) function, glucose regulation and autoimmune disorders through the production of many mediators, among which is Orexin-A [44]. The main effects of bariatric surgery on the nervous system are attributable to the vagus nerve that is involved during bariatric surgery. Geronikolou et al. reported that bariatric surgery has a greater effect on both branches of the cardiac automatic nervous system, making it more beneficial for severe cardiovascular patients. Furthermore, bariatric surgery has positive effects on insulin resistance by decreasing it. Moreover, this intervention is able to increase vagal tone, improving cardiac function. Heinonen et al. reported that gastric banding and/or gastric bypass are effective for weight loss, but among the beneficial effects of bariatric surgery there are many other factors such as neurohormonal feedback loops. Data in the literature report that peptide hormones such as Adiponectin, orexins, and leptin might play a part in this regulation [44,45].

Heinonen et al. reported that in the stomach and intestine enterochromaffin-like cells display Orexin immunoreactivity. Kirchgessner and Liu reported that Orexins play a role in the gastric and intestinal phase of secretion [46]. In addition, many literature data report the role of Orexin-A and B in appetite control [47,48]. Furthermore, it is known that Orexin inhibited responses to CCK suggesting its role in modulating gut-to-brain signaling. Several studies report that Orexin-A levels are influenced by bariatric surgery independent of weight loss. As reported by Gupta et al., in the acute post-operative phase (after 1 day) and prior to any weight loss, some subjects demonstrate an increase in Orexin while others have decreased plasma Orexin levels. This raises the question as to what factors regulate these acute changes in Orexin during the acute post-operative phase [49]. On the contrary, Federico et al. reported that Orexin-A serum levels were comparable in obese patients before and after bariatric surgery [50]. Moreover, Cigdem et al. showed that laparoscopic gastric band application resulted not only in significant weight loss but also in decreased Orexin-A serum levels [51]. Amin et al. demonstrated a rise in Orexin and a decrease in leptin in a cohort of obese subjects affected by obstructive sleep apnea, after bariatric surgery, showing that metabolic changes were also occurring in the same time span and thus it was plausible that physiologic rather than anatomic changes may underlie the clinically significant improvement in obstructive sleep apnea as early as 11 days following bariatric surgery [52]. The regulation of Orexin and the involvement of the nervous system both depend on mechanical factors but also on chemical mediators and hormones that are in circulation after the intervention. It is also reported that bariatric surgery has an impact on AT. Sams et al. reported that bariatric surgery has numerous beneficial effects on glucose homeostasis through the regulation and modulation of insulin sensitivity. The inflammatory mechanism of insulin resistance involves an early phase of improved insulin sensitivity and a late phase of decreased inflammatory mediators. The authors reported that gastric bypass induces a rearrangement of gastrointestinal anatomy and an amelioration in glucose homeostasis [53,54]. Regarding the rearrangement of gastrointestinal anatomy, bariatric surgery induces significant changes in gut hormone production that contribute to ameliorate general health [51]. It is well known that obesity induces not only an alteration of many metabolic mediators alerting glucose and lipid profiles, but also induces an imbalance in the expression of pro- and anti-inflammatory adipokines such as Adiponectin. As reported by Salman et al. bariatric surgery induces a significant increase in serum Adiponectin levels and a significant decline in serum levels of leptin, resisitin, and pre-B cell enhancing factor/Nampt/visfatin, confirming the role of this technique in hormonal rearrangement [55].

Moreover, recent studies have reported that bariatric surgery is associated with a reduction in specific adipokines including leptin, chemerin, and PAI-1, whereas Adiponectin is raised; adaptations that could be indicative of improved fat mass and function [56,57]. After bariatric surgery, obese subjects have an increase in Adiponectin serum levels. This increase is associated not only with weight loss, but also with a decrease of inflammation reducing insulin resistance (Table 1 and Figure 2).

## 7. Conclusions

Adiponectin and Orexin-A are both involved in obesity and its correlated diseases. These proteins are strongly reduced in obese patients. Both Adiponectin and Orexin-A have beneficial metabolic effects, increasing glucose uptake and insulin sensitivity; on the other hand, they are anti-inflammatory mediators. After bariatric surgery, Adiponectin and Orexin-A levels are increased and this is attributable not only to weight loss, but also to a rearrangement of metabolic and inflammatory mediators. The restriction of food intake or a combination of a restriction and malabsorption induced by bariatric surgery are, however, not sufficient to explain the profound clinical effects of bariatric surgery, suggesting the involvement of neuro-hormonal feedback loops and also of mediators such as Adiponectin and Orexin-A. Further studies are needed to clarify the molecular mechanism underlying this regulation.

## Figures and Tables

**Figure 1 jcm-09-03327-f001:**
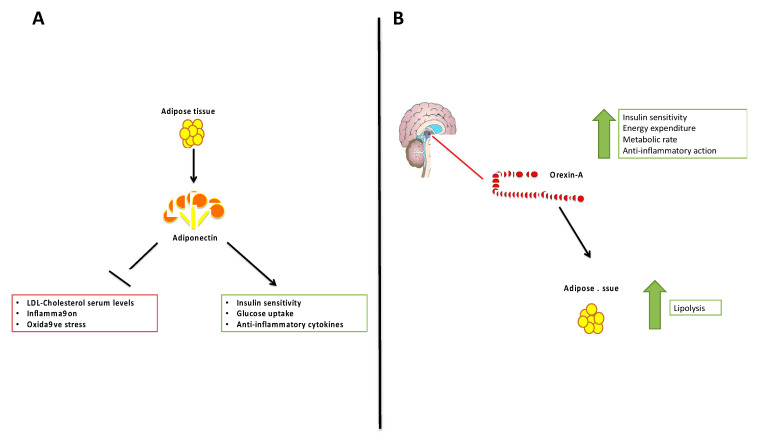
The principal beneficial effects of Adiponectin (panel (**A**)) and of Orexin-A (panel (**B**)).

**Figure 2 jcm-09-03327-f002:**
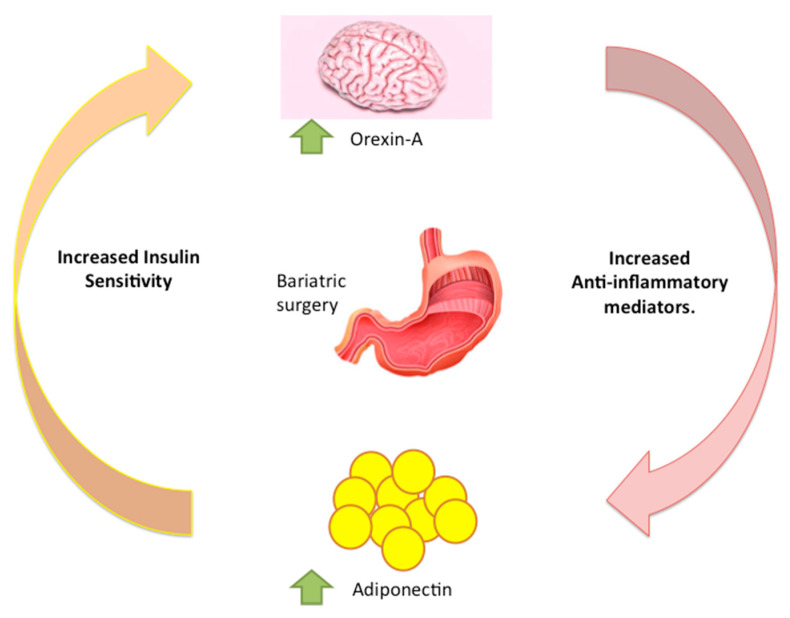
Adiponectin and Orexin-A levels are increased after bariatric surgery independently of weight loss showing the involvement of neuro-hormonal feedback loops.

**Table 1 jcm-09-03327-t001:** The main effects of bariatric surgery on adipose tissue and the central nervous system.

Bariatric Surgery	
Adipose Tissue	Central Nervous System
Increased Adiponectin	Increased Orexin-A
Decreased Leptin	Increased Vagal Tone
Decreased Insulin Resistance	Orexin-A Controls Appetite

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
