# Peer review of "The Metabolic Rearrangements of Bariatric Surgery: Focus on Orexin-A and the Adiponectin System"

_jcm, 2020, doi:10.3390/jcm9103327_

Round 1

Reviewer 1 Report

This interesting systematic review is describing the role of Adiponectin and Orexin-A in obese patients before and after bariatric surgery.

Some minor issues need to be addressed:

line 158: To my knowledge sleeve gastrectomy is the most common performed bariatric surgery worldwide (https://pubmed.ncbi.nlm.nih.gov/28405878/), not gastric bypass. Furthermore, gastric bypass is a reversible procedure!

lines 172-175: Guidelines for bariatric surgery may differ between countries. I would state that surgery is performed in subjects with a BMI > 35 kg/m2 with or without obesity-associated comorbidities.

line 178: I would not say that gastric bypass is more invasive than sleeve gastrectomy, since you don't remove any stomach or small intestine - you just rearrange. On the other side you remove 80% of stomach in sleeve gastrectomy, which is not reversible.

Author Response

This interesting systematic review is describing the role of Adiponectin and Orexin-A in obese patients before and after bariatric surgery.

Some minor issues need to be addressed:

line 158: To my knowledge sleeve gastrectomy is the most common performed bariatric surgery worldwide (https://pubmed.ncbi.nlm.nih.gov/28405878/), not gastric bypass. Furthermore, gastric bypass is a reversible procedure!

We thank the reviewer for this suggestion. In the revised manuscript, we reported:

“In addition, various bariatric surgery techniques use both procedures. Among these, gastric bypass is a reversible technique, decreasing the amount of food intake and also the absorption of nutrients. The most common performed bariatric surgery worldwide is sleeve gastrectomy as reported by Angrisani et al., 2014. This procedure consists in removing about 80% of the stomach, leaving a long, tube-like pouch. In addition, an adjustable gastric band (AGB) is another surgery technique.”

lines 172-175: Guidelines for bariatric surgery may differ between countries. I would state that surgery is performed in subjects with a BMI > 35 kg/m2 with or without obesity-associated comorbidities.

We thank the reviewer for this suggestion. In the revised manuscript, we reported:

“This surgical intervention is carried out on subjects with severe obesity. The current indications for bariatric surgery state that surgery is necessary when there is severe obesity and the potential reversibility of the clinical conditions. Classically, starting from the Consensus Conference of the American National Institute of Health (1991), Body Mass Index (BMI) is considered to be decisive, but it must be kept in mind that it has revealed important limits, not being able to highlight the distribution and breakdown of lipid accumulation in the form of somatic or visceral fat, a key factor in determining the metabolic syndrome; and also the different distribution of fat in relation to age, sex and race. For this reason, BMI is considered an important benchmark, but not the only one for establishing the indication for surgery. At the same time, BMI, also considered in its historical dimension as the maximum value reached by the patient, makes it possible to give indications for bariatric surgery. Finally, BMI is evaluated, together with metabolic, functional and psychological parameters, always in an overall balance between risks and benefits, in patients with BMI> 40 kg/m2, in the absence of any other comorbidities; or with BMI> 35 kg/m2, in the presence of comorbidities among those classically considered as associated with obesity, including type 2 diabetes mellitus (T2DM) resistant to medical treatment [38; Linea Guida di chirurgia dell’obesita, Ed.2016]”.

 line 178: I would not say that gastric bypass is more invasive than sleeve gastrectomy, since you don't remove any stomach or small intestine - you just rearrange. On the other side you remove 80% of stomach in sleeve gastrectomy, which is not reversible.

We thank the reviewer for this suggestion and in the text, we eliminated this incorrect sentence.

Reviewer 2 Report

The aim of the present review is to summarize the involvement of the autonomic nervous system in bariatric surgery. The topic is of particular interest in the field of bariatric surgery and the review may indeed represent a useful working resource for future research. However, some major concerns need to be addressed by the authors. 

Comments:

1. It is not clear why the authors decided to focus on adiponectin and orexin-A. 

2. The title is not appropriated since the authors do not focus on the general involvement of the autonomic nervous system but only in two related peptides. The manuscript would be more interesting if the authors could discuss more about the autonomic nervous system involvement on the bariatric surgery clinical results.

3. Figure 1 and 2 should be put together as Fig1A and Fig1B. There is no need for two figures with so little information.

4.The bariatric surgery section should be reorganized. 

    • Page 4, lines 154-155. The authors described when bariatric surgery is used. So, they should also include here the criteria for bariatric surgery.
    • The way that the criteria for bariatric surgery is written isn’t clear (page 5-lines 173-175).  It should be reformulated for something like: BMI of 40 or higher, or have a BMI between 35 and 39.5 and an obesity-related condition.
    • Page 4, lines 158-188. References are missing.
    • The authors introduce the sleeve gastrectomy at line 160, then they introduce the gastric band. At line 164, they return to the sleeve gastrectomy. The sleeve gastrectomy information should be put together.
    • Page 4, line 165-166. Isn’t this information about ghrelin not similar to other bariatric surgeries, such as gastric bypass?
    • Page 4, lines 156-157. The authors describe the malabsorptive surgeries but after that they ignore them. There is no information regarding BPD and DS surgeries in the bariatric surgery section.
    • Page 5, line 178. The authors describe gastric bypass as the most invasive bariatric surgery. What about malabsorptive surgeries? 
    • Page 7, lines 193-194- this sentence should be rephrased. 

5. For the better understanding of the section “Effects of bariatric surgery on Adiponectin and Orexin-A”, a table containing this information should be included. The results should be organized by bariatric surgery type and follow up after surgery.

6. Other important neuro-hormonal feedback loops in the context of obesity/bariatric surgery should be considered and included in the manuscript.

7. Page 5, line 210. Both peptides were discovered more than 20 years ago. I do not think that they should be considered “newly discovered”.

8. There is repeated information in some sections: Page 5 lines 170-171 = Page 5 lines 183-184; Page 6 lines 236-237 = Page 6 lines 238-239.

9. English should be revised. 

Author Response

The aim of the present review is to summarize the involvement of the autonomic nervous system in bariatric surgery. The topic is of particular interest in the field of bariatric surgery and the review may indeed represent a useful working resource for future research. However, some major concerns need to be addressed by the authors. 

Comments:

  1. It is not clear why the authors decided to focus on adiponectin and orexin-A. 

We thank the reviewer for this observation.

We decided to focus on adiponectin and orexin-A because these peptides have numerous common beneficial effects and are strongly involved in obesity development and resolution, and also because bariatric surgery induces a rearrangement in metabolic and hormone pathways. For these reasons we wanted to explore the role of these two hormones in bariatric surgery, being the subject of the study by our group also in other types of metabolic-functional diseases. Indeed, in the introduction, we added:

“Bariatric surgery is necessary when there is severe obesity and this technique is able to induce an improvement or resolution of many obese related conditions and to improve quality of life, strongly reducing weight loss and inducing a rearrangement of metabolic and hormone pathways. From data in the literature, it is clear that bariatric surgery is capable not only of acting from a mechanical-anatomical point of view, reducing the size of the stomach and therefore the intake of food, but also improving some metabolic parameters such as the production of adipocytokines and hypothalamic peptides. There is a strong metabolic interconnection between the central nervous system, the digestive system and adipose tissue”.

  1. The title is not appropriated since the authors do not focus on the general involvement of the autonomic nervous system but only in two related peptides. The manuscript would be more interesting if the authors could discuss more about the autonomic nervous system involvement on the bariatric surgery clinical results.

We changed the title to:

“The metabolic rearrangements of bariatric surgery: focus on orexin-A and the adiponectin system”

  1. Figure 1 and 2 should be put together as Fig1A and Fig1B. There is no need for two figures with so little information.

We combined the figures as Fig 1A and B, as the reviewer suggested.

4.The bariatric surgery section should be reorganized. 

  • Page 4, lines 154-155. The authors described when bariatric surgery is used. So, they should also include here the criteria for bariatric surgery.

We thank the reviewer for this suggestion and in the manuscript, we reported:

“The current indications for bariatric surgery refer to the severity of obesity and the potential reversibility of the clinical conditions. The evaluation of Body Mass Index (BMI) is considered as a marker for the indication for surgery. In particular, surgery is performed on subjects with a BMI> 40 kg/m2, in the absence of any other comorbidities and with a BMI > 35 kg/m2 with obesity-associated comorbidities [38; Linea Guida di chirurgia dell’obesita, Ed.2016].”

  • The way that the criteria for bariatric surgery is written isn’t clear (page 5-lines 173-175).  It should be reformulated for something like: BMI of 40 or higher, or have a BMI between 35 and 39.5 and an obesity-related condition

We thank the reviewer for this suggestion and in the manuscript, we reported:

“This surgical intervention is carried out on subjects with severe obesity. The current indications for bariatric surgery state that surgery is necessary when there is severe obesity and the potential reversibility of the clinical conditions. Classically, starting from the Consensus Conference of the American National Institute of Health (1991), the Body Mass Index (BMI) is considered to be decisive, but it must be kept in mind that it has revealed important limits, not being able to highlight the distribution and breakdown of lipid accumulation in the form of somatic or visceral fat, a key factor in determining the metabolic syndrome; and also the different distribution of fat in relation to age, sex and race. For this reason, BMI is considered an important benchmark, but not the only one for establishing the indication for surgery. At the same time, BMI, also considered in its historical dimension as the maximum value reached by the patient, makes it possible to give indications for bariatric surgery. Finally, BMI is evaluated, together with metabolic, functional and psychological parameters, always in an overall balance between risks and benefits, in patients with BMI> 40 kg/m2, in the absence of any other comorbidities; or with BMI> 35 kg/m2, in the presence of comorbidities among those classically considered as associated with obesity, including type 2 diabetes mellitus (T2DM) resistant to medical treatment [38; Linea Guida di chirurgia dell’obesita, Ed.2016].”

  • Page 4, lines 158-188. References are missing.

We added them.

  • The authors introduce the sleeve gastrectomy at line 160, then they introduce the gastric band. At line 164, they return to the sleeve gastrectomy. The sleeve gastrectomy information should be put together.

We thank the reviewer and we reported in the text:

“Bariatric surgery consists in some procedures limiting food intake and other procedures reducing the body's ability to absorb nutrients [36]. In addition, various bariatric surgery techniques use both procedures. Among these, the gastric bypass is a reversible technique, decreasing the amount of food intake and also the absorption of nutrients. The most common performed bariatric surgery worldwide is sleeve gastrectomy as reported by Angrisani et al., 2014. This procedure consists in removing about 80% of the stomach, leaving a long, tube-like pouch. Generally, sleeve gastrectomy induces notable weight loss, because it reduces stomach size and limits food intake. It also produces less of the appetite-regulating hormone ghrelin, which may lessen the desire to eat []. In addition, an adjustable gastric band (AGB) is another surgery technique. It has advantages including significant weight loss and it is an adjustable gastric band that is an inflatable silicone device placed around the top portion of the stomach to treat obesity, intended to decrease food consumption [angrisani et al.].”

  • Page 4, line 165-166. Isn’t this information about ghrelin not similar to other bariatric surgeries, such as gastric bypass?

Yes, for these reasons in the text we reported:

“Generally, sleeve gastrectomy induces notable weight loss, because it reduces stomach size and limits food intake. It also produces less of the appetite-regulating hormone ghrelin, which may lessen the desire to eat. In addition, ghrelin reduction is induced both by gastric bypass and sleeve gastrectomy [Svane at al., Postprandial Nutrient Handling and Gastrointestinal Hormone Secretion After Roux-en-Y Gastric Bypass vs Sleeve Gastrectomy. Original Research Full Report: Clinical—Alimentary Tract| Volume 156, ISSUE 6, P1627-1641.e1, May 01, 2019]”.

  • Page 4, lines 156-157. The authors describe the malabsorptive surgeries but after that they ignore them. There is no information regarding BPD and DS surgeries in the bariatric surgery section.

We thank the reviewer for this observation, in the text we reported:

190-195: Bariatric surgery consists in some procedures limiting food intake and other procedures reducing the body's ability to absorb nutrients. Malabsorptive bariatric procedures divert the flow of bile and pancreatic enzymes from food and therefore limit the digestion and absorption of nutrients, resulting in reduced calorie intake and subsequent weight loss. Essential micronutrients such as vitamins and trace elements are also absorbed to a lesser extent, potentially leading to severe side effects [36].

Line 220-232: Another procedure is a biliopancreatic diversion with duodenal switch (BPD/DS). It is a complex procedure that tackles weight loss in 3 different ways. First, a sleeve gastrectomy is performed. For this, a large portion of the stomach is removed with a stapling instrument, leaving a narrow tube, or sleeve, from the top to near the bottom of the stomach. The second part of the procedure reroutes food away from the upper part of the small intestine, which is the natural path of digestion. This cuts back on how many calories and nutrients the body is able to absorb. The small intestine is divided and a connection is made near the end of the small intestine. The third part of the BPD/DS procedure changes the normal way that bile and digestive juices break down food. This cuts back on how many calories you absorb, causing still more weight loss. One end of the small intestine is connected to the duodenum, near the bottom of the stomach [Pappadia et al., “BPD and BPD-DS Concerns and Results.” Advanced Bariatric and Metabolic Surgery February 2012. DOI: 10.5772/32041].

  • Page 5, line 178. The authors describe gastric bypass as the most invasive bariatric surgery. What about malabsorptive surgeries? 

We thank the reviewer for this suggestion and we eliminated this incorrect sentence.

  • Page 7, lines 193-194- this sentence should be rephrased. 

We corrected and reported in the manuscript as:

“Moreover, bariatric surgery, the central nervous system and AT influence each other through the production of neuropeptides such as orexin-A and adipocytokines such as adiponectin as suggested by several studies.”

  1. For the better understanding of the section “Effects of bariatric surgery on Adiponectin and Orexin-A”, a table containing this information should be included. The results should be organized by bariatric surgery type and follow up after surgery.

We added a table in the manuscript.

  1. Other important neuro-hormonal feedback loops in the context of obesity/bariatric surgery should be considered and included in the manuscript.

We thank the reviewer for this suggestion, but we believe that the focus of our manuscript from the point of view of the central nervous system, is the orexin-A system

  1. Page 5, line 210. Both peptides were discovered more than 20 years ago. I do not think that they should be considered “newly discovered”.

We corrected it as:

“Data in the literature report that peptide hormones such as Adiponectin, orexins, and leptin might play a part in this regulation”

  1. There is repeated information in some sections: Page 5 lines 170-171 = Page 5 lines 183-184; Page 6 lines 236-237 = Page 6 lines 238-239.

We corrected this repeated information in the text.

  1. English should be revised. 

Round 2

Reviewer 2 Report

The authors answered all my questions and revised the manuscript as sugested.